# HDAC Inhibitors Enhance Efficacy of the Oncolytic Adenoviruses Ad∆∆ and Ad-3∆-A20T in Pancreatic and Triple-Negative Breast Cancer Models

**DOI:** 10.3390/v14051006

**Published:** 2022-05-09

**Authors:** María Del Carmen Rodríguez Rodríguez, Inés García Rodríguez, Callum Nattress, Ahad Qureshi, Gunnel Halldén

**Affiliations:** 1Centre for Biomarkers and Biotherapeutics, Barts Cancer Institute, Queen Mary University of London, London EC1M 6BQ, UK; c.rodriguez@qmul.ac.uk (M.D.C.R.R.); ahad.qureshi@qmul.ac.uk (A.Q.); 2OrganoVIR Labs, Department of Medical Microbiology, Amsterdam Institute for Infection and Immunity, Amsterdam UMC Location University of Amsterdam, Meibergdreef 9, 1105 AZ Amsterdam, The Netherlands; inegarciarguez@gmail.com; 3Cell Communication Lab, Department of Oncology, University College London Cancer Institute, London WC1E 6DD, UK; callum.nattress.19@ucl.ac.uk

**Keywords:** oncolytic adenovirus, epigenetic, histone deacetylase inhibitor, HDACi, TNBC, PDAC

## Abstract

The prognosis for triple-negative breast cancer (TNBC) and pancreatic ductal adenocarcinoma (PDAC) is dismal. TNBC and PDAC are highly aggressive cancers with few treatment options and a potential for rapid resistance to standard-of-care chemotherapeutics. Oncolytic adenoviruses (OAds) represent a promising tumour-selective strategy that can overcome treatment resistance and eliminate cancer cells by lysis and host immune activation. We demonstrate that histone deacetylase inhibitors (HDACi) potently enhanced the cancer-cell killing of our OAds, Ad∆∆ and Ad-3∆-A20T in TNBC and PDAC preclinical models. In the TNBC cell lines MDA-MB-436, SUM159 and CAL51, cell killing, viral uptake and replication were increased when treated with sublethal doses of the Class-I-selective HDACis Scriptaid, Romidepsin and MS-275. The pan-HDACi, TSA efficiently improved OAd efficacy, both in vitro and in SUM159 xenograft models in vivo. Cell killing and Ad∆∆ replication was also significantly increased in five PDAC cell lines when pre-treated with TSA. Efficacy was dependent on treatment time and dose, and on the specific genetic alterations in each cell line. Expression of the cancer specific αvß6-integrin supported higher viral uptake of the integrin-retargeted Ad-3∆-A20T in combination with Scriptaid. In conclusion, we demonstrate that inhibition of specific HDACs is a potential means to enhance OAd activity, supporting clinical translation.

## 1. Introduction

Triple negative breast cancer (TNBC) is the most aggressive subtype of breast cancer (BCa), accounting for 15% of all BCa with no curative treatment in the late stages of disease, affecting mostly young premenopausal women [1]. TNBC is characterized by the lack of oestrogen (ER) and progesterone receptors (PR), and the human epidermal growth factor receptor 2 (HER2). Thus, patients do not respond to the standard-of-care hormonal therapies, including the ER antagonist Tamoxifen, the anti-HER2 monoclonal antibody Trastuzumab (Herceptin) or the dual HER2/EGFR tyrosine kinase inhibitor Lapatinib. Pancreatic ductal adenocarcinomas (PDAC) have a similar dismal prognosis and are the fourth leading cause of cancer-related death globally with a 5-year survival rate of less than 9% [2]. Major reasons for the poor survival are late presentation of symptoms and rapid development of resistance to all therapeutics [3,4]. For both TNBC and PDAC, chemotherapy along with surgery are currently the only treatment options. Novel therapeutics with different mechanisms of action are urgently needed. A promising strategy with reported clinical safety is the application of replication-selective oncolytic adenoviruses (OAd) that have native tropism for adenocarcinomas, kill tumour cells by lysis and immune activation, and synergise with chemotherapeutics to overcome drug resistance [5].

We previously developed two potent cancer-selective oncolytic adenoviruses, Ad∆∆ and Ad-3∆-A20T, with the E1ACR2 region and the E1B19K gene deleted [6,7,8,9,10]. E1ACR2 is essential for pRb binding and inactivation to promote S-phase entry and support viral propagation in normal cells. The deletion restricts viral replication to cancer cells with deregulated cell cycle. E1B19K is an anti-apoptotic functional Bcl-2 homolog that blocks the release and binding of death ligands to Bax and Bak proteins [8]. E1B19K-deleted mutants sensitize cells to apoptosis in combination with apoptosis-inducing chemotherapeutic drugs [8,11]. Ad∆∆ potently eliminates prostate cancer and PDAC cells in preclinical models with superior efficacy in combination with DNA-damaging drugs compared to the parental virus HAdV5 [6,7,9,12]. The Ad-3∆-A20T mutant was generated to improve on systemic delivery for elimination of distant metastasis by retargeting to tumour-specific αvß6-integrins by inserting a 20 amino acid peptide from the Foot-and-Mouth-Disease-Virus (A20FMDV) in the fibre knob and de-target blood factor binding [9,10,13]. The αvß6-integrin is expressed at high levels in invasive breast cancer and other solid tumours including PDAC, but not in adult normal cells [9,14,15]. We previously showed that Ad-3∆-A20T reached αvß6-integrin-expressing PDAC tumours in higher quantities than the parental Ad∆∆ after systemic delivery [9,10]. Moreover, the retargeting could potentially inhibit the binding of the fibre to coagulation factors that usually lead to liver toxicity and/or rapid elimination of the circulating virus.

The viral life-cycle in normal cells is dependent on the acetylation and deacetylation of Lysine residues in histone tails and in cellular and viral proteins [16,17,18,19]. These processes are regulated by histone acetylases (HATs), histone deacetylases (HDACs) and viral E1A-binding to the p300/CBP histone acetyltransferase, resulting in transcriptional activation and repression of specific genes [17,20,21]. Epigenetic alterations also play important roles in cancer development and progression, with specific epigenetic profiles dependent on tumour type. For example, HDAC enzymes from Class I (HDAC1, 2, 3 and 8) have been reported to be overexpressed in breast cancer [22,23]. HDAC2 and 3 are frequently expressed at high levels in hormone-receptor negative tumours, while aberrant HDAC1 expression is common in hormone-receptor positive tumours. Inhibition of numerous HDACs with targeted drugs in combination with current therapeutics have shown promise in the clinic [24]. These include the Class I specific inhibitors Scriptaid, Romidepsin and MS-275, and the pan-inhibitor Trichostatin A (TSA) that targets all isoforms of the zinc-dependent HDAC classes [25]. Romidepsin is an approved anti-cancer drug and was reported to enhance adenoviral transgene expression and infection in melanomas [26,27]. Scriptaid was demonstrated to enhance oncolysis of the OAd mutant Ad∆24-RGD in glioma explants [28]. TSA greatly reduced cell viability in cisplatin-resistant ovarian cells in combination with cisplatin and Ad5wt or the OAd mutant *dl*24 [29]. Other HDACis such as Vorinostat showed increased anti-cancer activity in combination with Ad∆24, although the drug attenuated viral replication [30]. 

Rearrangement of histone tail marks (mainly acetylation) at host cell promoters are essential processes for effective adenoviral propagation in normal cells [16,17,18,19]. We speculated that timely inhibition of HDAC activity could enhance viral potency by increasing acetylation at essential promoters also in cancer cells and/or acetylation of viral factors. Here we report the effects of four HDACis on viral uptake, replication and cell killing of Ad∆∆ and Ad-3∆-A20T mutants in six TNBC and five PDAC cell lines. We demonstrate increased oncolytic activity in combination with the pan-HDACi TSA and with the more selective HDACi Scriptaid in the TNBC cells MDA-MB-436, CAL51 and SUM159. The combination of TSA and Ad∆∆ effectively suppressed the growth of SUM159 xenografts in athymic mice and prolonged the time to tumour progression. The addition of Scriptaid in combination with the retargeted Ad-3∆-A20T further improved on anti-cancer efficacy. Our findings indicate that both Ad∆∆ and Ad-3∆-A20T may be developed into clinical therapeutics in combination with selective HDACis to improve on outcomes for TNBC and PDAC patients.

## 2. Materials and Methods

Cell lines and culture conditions. Human triple-negative breast cancer (TNBC) and pancreatic ductal adenocarcinoma (PDAC) cell lines were used in the study, with genetic alterations listed in Appendix A). TNBC: BT549, HCC1143, SUM149, SUM159 (primary tumours); CAL51, MDA-MB-436 (pleural effusion metastasis). PDAC: MiaPaCa2, Panc04.03, BxPC3, PT45 (primary tumours), Suit2 (liver metastasis). All TNBC cells were obtained from our collaborator (Prof. S. Linardopoulos, Institute of Cancer Research, London, UK), and the PDAC cells were purchased from ATCC except for PT45, which was a kind gift from Prof H. Kalthoff (Comprehensive Cancer Centre, Campus Kiel, Germany). Human embryonic kidney cells, HEK293 (ATCC), were used for viral replication assays. Cells were grown in Dulbecco’s modified Eagle’s medium (DMEM; Gibco), supplemented with 10% foetal bovine serum (FBS; Thermo Fisher), and 1% penicillin/streptomycin (P/S; Thermo Fisher Scientific, Waltham, MA, USA) except SUM159 and SUM149 cells, which were grown in DMEM/Nutrient Mixture F-12 Ham (Gibco) supplemented with 5% FBS and 1% P/S. HCC1143 cells were grown in Roswell Park Memorial Institute medium (RPMI; Gibco), supplemented with 10% FBS and 1% P/S. All cell lines were grown at 37 °C in an atmosphere of 5%CO_2_.

### 2.1. Viruses and Inhibitors

The following viruses and mutants were used in the study: human wild-type adenovirus type 5 (Ad5wt), Ad∆∆ (Ad5 deleted in E1ACR2 and E1B19K; [6]), Ad5-3∆-A20T (Ad∆∆ retargeted to αvß6-integrin; [9]), and the EGFP-expressing Ad5-mutants; AdwtGFP and AdFMDVGFP [9]. The Class I histone deacetylase inhibitors (HDACi) Scriptaid (Sigma-Aldrich, Dorset, UK), MS275 (Entinostat; Selleckchem, Radnor, PA, USA), Romidepsin (FK228; Selleckchem) and the pan-HDACi Trichostatin A (TSA; Abcam, Cambridge, UK) were stored in DMSO as 1 mM stock solutions and diluted in media prior to the studies. 

### 2.2. Cell Viability Assay and Synergistic Interactions

Cells were seeded in 96-well plates (1 × 10^4^ cells/well) in 10% FBS-DMEM and infected with the respective virus after 24 h in 2% FBS-DMEM, and HDACis were added simultaneously (=), 24 h after infection (+24 h) or 24 h before infection (−24 h). Cells were analysed for cell viability 6 d after infection by the MTS assay [3-(4,5-dimethylthiazol-2-yI)-5-(3-carboxymethoxyphe-nyl)-2-(4-sulfophenyl)-2H-tetrazolium] (Promega, Madison, WI, USA) according to the manufacturer’s instructions. Dose-response curves were generated to calculate the EC_50_-values and to determine the dose of virus killing 50% of cells under each condition, using untreated cells or cells treated with drug as controls. Each data point was generated from 3–6 samples and repeated at least 3 times, as previously described [6,11]. Synergistic conditions were determined using fixed concentrations of virus and drug killing < 30% of cells alone. Cells were treated as described for the dose-response assays and the percentages of dead cell determined by the MTS assay 3 or 6 d after infection. Each data point was generated from triplicate samples and repeated at least 3 times.

### 2.3. Viral Replication Assay by Tissue Culture Infectious Dose (TCID_50_)

Cells were seeded in 6-well plates (1 × 10^5^ cells/well), infected with the virus at 100 particles per cell (ppc) and harvested 24–120 h post-infection. Fixed concentrations of HDACis were added at the respective times according to the conditions of the experiments. Cells and media were harvested, and viruses were released by three freeze-thaw cycles (37 °C and N_2(l)_). Viral replication was quantified by the TCID_50_ limiting dilution method on JH293 cells (HEK293 subclone). Cytopathic effects were determined in each well 8–10 days after infection in duplicate samples. Data were averaged and expressed as plaque-forming units (pfu) per ml, as previously described [6,9].

### 2.4. Immunoblotting

Cells were seeded in 6-well plates (3 × 10^5^ cells/well) and infected with the respective virus for 48 h. Fixed concentrations of inhibitor drugs were added at indicated timepoints. After 48 h, cells were lysed with RIPA buffer (50 mM Tris-HCl pH 8.0, 150 mM NaCl, 1% Triton X-100, 0.5% sodium deoxycholate, 0.1% sodium dodecyl sulphate; SDS; Sigma-Aldrich) containing protease inhibitors (Roche). Proteins were quantified using the bicinchoninic acid (BCA) assay using a BSA standard curve (Thermo Fisher Scientific). Protein samples were denatured in Laemmli buffer (0.5 M Tris-HCl pH6.8, 10% Glycerol, 2% SDS, 0.25% bromophenol blue, 10% β-mercaptoethanol). Protein samples (18–40 μg/lane) were separated on 10 or 12% polyacrylamide gel electrophoresis (SDS-PAGE) and transferred to polyvinylidene (PVDF) membranes (Millipore, Burlington, MA, USA). Proteins were detected by the following antibodies: mouse anti-Ad5 E1A (1:1000; Invitrogen, Waltham, MA, USA), rabbit anti-Ad5 coat protein (1:1000; Abcam, Cambridge, UK), rabbit anti-αvß6 integrin (1:500; Abcam) and mouse anti-PCNA (1:1000; Santa Cruz Biotechnology, Dallas, TX, USA). Detection was carried out using goat anti-mouse and goat anti-rabbit (1:2000) horseradish peroxidase-conjugated secondary antibodies (Dako, Copenhagen, Denmark) and Enhanced Chemiluminescence (ECL) reagent (Sigma-Aldrich, Dorset, UK) and visualized by Chemidoc imaging system (Amersham Imager 600, Amersham, UK).

### 2.5. Flow Cytometric Analysis

Cells were seeded in 6-well plates (2 × 10^5^ cells/well) and infected 24 h later at 100 ppc with the respective GFP expressing viruses; Ad5wtGFP and AdFMDVA20T. Fixed doses of Scriptaid and TSA was added at indicated times. Cells were harvested 48 h post infection and resuspended in FACS Buffer (0.5 g BSA, 500 mL 10% FBS DMEM). The infection study was performed on an LSRFortessa cytometer using a blue laser, acquiring 10,000 events per duplicate sample. Each experiment was repeated 2–3 times. Data was analysed using the FACSDiva software and expressed as the percentage of cell expressing GFP protein. 

### 2.6. In Vivo Tumour Growth

Cancer cells were inoculated subcutaneously in one flank of CD*nu/nu* athymic mice (Charles River, UK) with Suit2 or SUM159 cells (1 × 10^6^ cells/200μL) in sterile PBS. Treatments were initiated when tumours were 80–100μL by intratumoural administration (i.t) of adenoviral mutants. Animals with Suit2 xenografts were administered Ad∆∆ (1 × 10^9^ vp/50 µL) i.t. on day 2, 5 and 9, and TSA (20 µg/200 µL) on day 1, 4, and 8, intraperitoneally (i.p). Animals with SUM159 xenografts were injected with Ad∆∆ (1 × 10^10^ vp/50 µL) i.t. on day 1, 3 and 7, and TSA (20 µg/200 µL) administered i.p. on day 2, 4 and 8. Tumour growth, progression and animal weight were followed until tumours reached 800 µL or until clinical signs were observed (according to UK Home Office Regulations). Tumour volumes were estimated twice weekly: volume = (length × width^2^ × π)/6. Survival analysis was performed according to the method of Kaplan–Meier (log rank test for statistical significance). At the end of the study. Tumours were harvested and fixed in 4% formaldehyde. The fixed tissues were sectioned and processed for histopathology with H/E and for IHC by staining for E1A (1:200; AutogenBioclear, Wiltshire, UK) followed by detection using HRP-conjugated secondary antibodies (Dako). 

### 2.7. Statistical Analysis

All graphs and dose-response curves were generated with GraphPad Prism version 8.0 (GraphPad Software, San Diego, CA, USA). Means between two samples were compared by unpaired Student *t*-test; *p*-values < 0.05 were considered statistically significant. Figures show the mean of representative and independent repeats ± Standard Deviation (SD).

## 3. Results

### 3.1. The Pan-HDACi TSA Sensitises TNBC and PDAC Cell Lines to Ad∆∆-Mediated Cell Killing

The effect on Ad∆∆-mediated cell killing by the pan-HDACi TSA was investigated in six TNBC and five PDAC cell lines. The cells were simultaneously infected with increasing doses of Ad∆∆ (from 0.0002 ppc to 20,000 ppc) and fixed concentrations of TSA (0.1–0.5 µM) that did not significantly contribute to cell killing alone (<30% in all cell lines). Cell viability was determined from Ad∆∆ dose-response curves 6 days later (TNBC and PDAC: Appendix A) and expressed as ratios of EC_50_-values for combination treatment relative to virus alone (Figure 1A,B). In the TNBC cells CAL51, MDA-MB-436 and SUM159, TSA significantly (*p* < 0.01) decreased Ad∆∆ EC_50_-values. Similarly, TSA treatment of the PDAC cells MiaPaCa2, Panc04.03, Suit2 and BxPC3 resulted in significantly (*p* < 0.01) reduced Ad∆∆ EC_50_ values. The greatest sensitisation was observed in SUM159 and MDA-MB-436 cells with 12-fold and 2.5-fold and in Suit2 and MiaPaCa2 cells with 6-fold and 4-fold decreases in EC_50_ values, respectively. 

To determine if the increased cell killing was caused by increased viral uptake, cells were infected with Ad5wtEGFP ± TSA, followed by quantification of EGFP expression (Figure 1C,D). The TNBC cells BT549 and SUM149 were poorly infectible (<10% of cells), and MDA-MB-436 showed the highest levels of viral uptake (>50%) (Figure 1C). A significant and dose-dependent increase in viral uptake was observed only in CAL51 cells in combination with TSA. In contrast, TSA did not increase viral uptake in other TNBC cell lines at the tested doses (0.5 µM, 1.0 µM). Similar findings were seen in the PDAC cells with no significant increase of infection at any TSA concentration except a small increase in Suit2 cells with the lower dose of 0.5 µM TSA (Figure 1D). 

These data show that, in eight out of eleven cell lines, Ad∆∆-mediated cell killing was greatly increased in response to simultaneous treatment with TSA and appeared to be independent of the level of viral uptake, with only minor increases in the CAL51 and Suit2 cells. 

### 3.2. Effects on Viral Replication Is Dependent on Cell Line and Time of TSA Addition

Three of the PDAC cell lines, Suit2, BxPC3 and Panc04.03, that showed the greatest increases in Ad∆∆-mediated cell killing and minimal effects on infection in response to TSA were assessed for changes in viral replication. TSA did not enhance viral replication when added simultaneously at a sublethal dose (0.25 µM) and decreased replication at the higher concentration of 0.5 µM (Figure 2A; left panel). In contrast, when TSA (0.25 µM) was added 24 h prior to Ad∆∆-infection, replication was significantly increased in Suit2 and BxPC3 cells (Figure 2A; right panel). Replication in Panc04.03 cells was not stimulated under any conditions. Interestingly, in the TNBC cells that showed the greatest effects on Ad∆∆-induced cell killing with simultaneous addition of TSA (CAL51, MDA-MB-436 and SUM159), no significant increases in virion production were detected (Figure 2B). Trends towards less viral production were noted in CAL51 cells, while production was promoted in MDA-MB-436 cells after simultaneous additions. Increased replication was seen in CAL51 cells when TSA was added 24 h after infection (Figure 2B; left panel). In contrast, preincubation with TSA appeared to decrease viral yields in all three cell lines (Figure 2B; right panel). Further analysis of replication rate showed significantly higher levels in CAL51 and MDA-MB-436 cells when TSA was added after infection and simultaneously, respectively (Figure 2C; left and middle panels). The greatest increase was from 24 to 48 h in CAL51 and from 48 to 72 h in MDA-MB-436. In SUM159, a significantly higher replication rate was noted first from 48 to 120 h when TSA was added after the virus (Figure 2C; right panel, Appendix A). As expected, there was a good correlation between the Ad∆∆ replication data and expression of the early E1A protein in all TNBC cell lines treated with TSA ( Figure 2B,C and Appendix A). E1A expression in the presence of TSA increased over time similar to levels in cells infected with Ad∆∆ alone.

In summary, viral replication was enhanced by adding a low dose of TSA, either simultaneously or after viral infection in CAL51, MDA-MB-436 and SUM159. In contrast, increased replication in PDAC cells was enhanced when cells were pre-treated with TSA. These finding demonstrate the positive interactions between OAds and HDACi and warrants further investigation in relation to timing, dose of drug and gene alterations.

### 3.3. TSA Promote Ad∆∆ Efficacy in SUM159 and Suit2 Tumour Xenografts In Vivo

To confirm the potent cell killing and/or replication observed in vitro, Suit2 and SUM159 xenografts were grown subcutaneously in athymic mice. In animals with SUM159 tumours, combination treatments with Ad∆∆ and TSA resulted in significant inhibition of tumour growth 15 days post treatment compared to single agent treated animals (Figure 2D and Appendix A). Time to tumour progression was prolonged, with 100% of animals showing only small tumours (150–300 µL) at the end of the study (29 days post treatment) (Figure 2E). In contrast, only 20% (Ad∆∆) and 40% (TSA) of single agent treated animals had tumours <400 µL. Tumours harvested from animals with Suit2 tumours showed potent early viral gene expression, both in the presence and absence of TSA up to 25 days post virus treatment (Appendix A). A trend towards higher E1A levels and viral spread was noted in the combination treated animals when TSA was administered 24 h prior to virus, although, differences could not be quantified under the current conditions.

### 3.4. Selective HDAC Inhibitors Promote Ad∆∆ Infection and Propagation

TSA is a pan-HDACi that inhibits all isoforms of the zinc-dependent HDAC classes I, II and IV [31]. We sought to identify a more selective HDACi that would recapitulate the positive effects of TSA on the viral life cycle and focused on the TNBC cell lines CAL51, MDA-MB-436 and SUM159 as representative models. We screened the cells using high affinity inhibitors of the Class I enzymes, Romidepsin, MS-275 and Scriptaid at concentrations that killed <20% of cells when administered alone. All three inhibitors supported viral infection (Figure 3A). Viral uptake was significantly increased in CAL51 with 0.25 µM Scriptaid when added simultaneously or 24 h post-infection, although to a lesser degree than with 0.25 µM TSA. In the poorly-infectible SUM159 cells, all inhibitors increased viral infection to significantly higher levels than with TSA, both when added simultaneously and after virus (Figure 3A). No major changes in viral uptake were detected in MDA-MB-436 cells with the Class I inhibitors.

The increased infectivity in the presence of inhibitors was reflected in enhanced Ad∆∆-mediated cell killing and replication under the same conditions (Figure 3B,C). Notably, Romidepsin, MS-275 and Scriptaid added 24 h after virus significantly increased replication 48 h post-infection in SUM159 (Figure 3B). Moreover, cell killing was synergistic in SUM159 with Scriptaid added 24 h post-infection (Figure 3C; left panel). Similarly, Ad∆∆ replication in MDA-MB-436 cells was enhanced by MS-275 added simultaneously and with Scriptaid and MS-275 added 24 h post-infection (Figure 3B), while cell killing was synergistic with Scriptaid and additive with MS275 under these conditions (Figure 3C; middle panel). In CAL51 cells, viral replication and cell killing was supported by the addition of HDACis, although no significant improvements were observed (Figure 3B,C; right panel).

All three Class I inhibitors promoted viral infection, replication and cell killing when added at low doses simultaneously and/or after viral infection. The response to Scriptaid appeared to be more potent, indicating a possible stimulatory effect on the virus by HDAC1 and 2, and potentially HDAC6 [32], which is also inhibited by TSA but not by other tested inhibitors. Scriptaid was therefore selected for further in-depth studies.

### 3.5. Combination Treatments with Ad∆∆ and Scriptaid Enhance Virus-Induced Cell Death and Replication in TNBC Cell Lines

With the focus on Scriptaid, we assessed changes in Ad∆∆ EC_50_ values in cells treated with and without a low dose of the inhibitor. Significant decreases in EC_50_ values were seen in both SUM159 (0.5 µM; 62–76%) and MDA-MB-436 (0.25 µM; 62–67%) cells in the presence of Scriptaid (Appendix A and Appendix A). In contrast, a lesser effect was noted in CAL51 cells with Scriptaid (0.25 µM; 26–27%). CAL51 cells had >10-fold greater sensitivity to Ad∆∆ alone compared to other cell lines, a likely reason for the lack of significant sensitization to Scriptaid. To explore effects on replication in the presence of Scriptaid, viral yields were determined over time from 24–120 h after infection. In SUM159 cells, a more than 1000-fold increase was observed from 24–120 h, and the addition of Scriptaid did not significantly enhance replication under any conditions but paralleled the observed increases with Ad∆∆ alone over time (Appendix A). In MDA-MB-436 cells, Ad∆∆ increased more than 10-fold from 24 to 96 h and plateaued between 96 and120 h, while in the presence of Scriptaid added simultaneously, replication was enhanced only after 96 h of infection (Appendix A). These data demonstrate that low doses of Scriptaid enhance Ad∆∆-mediated cell killing and support viral replication over time.

### 3.6. Infectivity of the αvß6-Integrin Targeted Mutant Ad5-3∆-A20T Is Enhanced by Scriptaid

The αvß6-integrin is selectively expressed in numerous solid cancers including BCa [14,15]. To assess the potential of our αvß6-integrin targeted Ad5-3∆-A20T OAd as an anti-cancer therapy in TNBC, viral uptake was explored using Ad-3∆-A20T-EGFP (same tropism as Ad5-3∆-A20T) and compared to Ad5wtGFP (same tropism as Ad∆∆) (Figure 4A). In SUM159, up to 12-fold increases were seen with Scriptaid added simultaneously with either virus (left panel). Similar trends were observed in CAL51 and MDA-MB-436 cells, with increased EGFP-expression after infection with Ad-3∆-A20T-EGFP in the presence of Scriptaid, although to a lesser degree than in SUM159 cells (Figure 4A; mid and right panels). Both the MDA-MB-436 and the CAL51 cells were significantly more infectable than SUM159, reaching 20–35% with the virus alone (0.5–1.0% in SUM159), possibly explaining the smaller increases in uptake with Scriptaid. Interestingly, we found that in all three tested TNBC cell lines there were trends towards higher levels of αvß6-integrin expression in the presence of Scriptaid (Figure 4B and Appendix A).

### 3.7. Scriptaid and TSA Enhance Ad5-3∆-A20T-Dependent Cell Killing and Replication in TNBC Cells

To evaluate whether the cell killing efficacy of the tumour-targeted Ad5-3∆-A20T could be enhanced by Scriptaid and TSA, similar to the observed effects on Ad∆∆, virus-mediated cell killing and propagation were investigated in SUM159 and MDA-MB436 cells. Sensitization to virus was observed in the presence of either HDACi (Figure 5A,B). The greatest relative decreases in EC_50_ values (up to 75%) were seen in SUM159 cells with Scriptaid added together with Ad5-3∆-A20T and in MDA-MB-436 cells with TSA (up to 80%). Viral replication was determined after the addition of a low dose of Scriptaid (0.1 µM) in cells simultaneously infected with Ad5-3∆-A20T (100 ppc). Significantly increased Ad5-3∆-A20T replication was observed after 48 h in both cell lines (Figure 5C). These findings demonstrate that the HDACi Scriptaid enhanced cell killing through increased uptake and replication of Ad5-3∆-A20T at low doses of the inhibitor.

## 4. Discussion

There are currently no curative therapeutics for late-stage TNBC and PDAC, two of the most aggressive and deadly adenocarcinomas globally [1,2]. There is an urgent need for novel improved therapies to treat these patients. Clinical trials with OAds in patients with various adenocarcinomas, including PDAC and BCa, have shown promising anti-tumour efficacy with proven biological activity and safety [33,34,35]. OAds act as superior sensitisers for chemodrug-resistant tumours through their unique cell killing mechanisms, a combination of cancer cell lysis and immunogenic cell killing [5,36]. However, certain DNA-damaging chemotherapeutics administered at high doses may also impede viral functions [11]. Therefore, we explored the potential of combining OAds with the more recently developed HDACis that cause less DNA damage. We found that several HDACis worked synergistically with our OAd mutants, Ad∆∆ and Ad-3∆-A20T, by specifically increasing cell killing efficacy and replication in human TNBC and PDAC cell models.

It has been established that adenoviruses remodel the epigenetic landscape in infected normal cells [16,17,18,19]. The deacetylation of differentiation-related genes and re-acetylation of promoters controlling S-phase entry are absolute requirements for adenovirus propagation. Furthermore, acetylation of the Rb protein and the viral E1A protein are also essential for efficient viral replication [18]. We hypothesised that timely inhibition of specific HDACs would amplify viral gene transcription and production of virions through maintenance of acetylation at critical promoters to enhance cancer cell lysis and intratumoural spread. To date, several HDACis have been evaluated in the clinic, including TSA, MS-275, Romidepsin and Scriptaid [32,37,38]. However, as monotherapies, the inhibitors had only low anti-tumour efficacy in solid tumours [24]. Our findings demonstrate that TSA greatly enhanced cell killing and replication of our OAds in both TNBC and PDAC models in vitro and in vivo. Previous studies showed that TSA in combination with the OAd *dl*24 sensitised cisplatin resistant ovarian A2780 cells to cisplatin and decreased cell viability [29]. It was suggested that the increased cell killing was TSA-mediated inhibition of HDAC1 and 2 that were upregulated in the resistant cells. TSA has also been reported to increase expression of membrane-receptors such as Coxsackie and Adenovirus Receptor (CAR) or integrins as demonstrated in bladder and lung cancer cell lines [39]. We did not examine expression levels of the native viral receptors since only minor increases in viral uptake was observed in CAL51 and Suit2 cells, but rather focused on the improved oncolysis. However, we observed a trend towards higher αvß6-integrin levels when TNBC cells were grown in the presence of Scriptaid, which was accompanied by increased uptake of the AdFMDV mutant in SUM159 cells. Most likely the inhibitor-mediated alterations of gene transcription facilitated viral protein expression and/or genome amplification as we observed an increased level of viral replication and significantly enhanced cell killing.

TSA is a pan-HDACi that inhibits all isoforms of the zinc-dependent HDAC classes I, II and IV [25]. To identify more specific HDACs for targeted therapies we explored whether the positive effects on the viral life cycle could be reproduced by selective Class I inhibitors. Scriptaid targets HDAC1, 2, and 8, and HDAC6 of Class IIb and has been developed as a potent HDACi with anti-tumour activity against several cancers such as endometrial, ovarian, colon or lung cancer [32,40]. Synergistic effects of Scriptaid in combination with the OAd Ad∆25-RGD was previously demonstrated in patient-derived glioblastoma cells [28]. Improved anti-tumour efficacy has also been reported with the potent HDAC2 inhibitor valproic acid (VPA) that supported Ad-replication in colon carcinoma [41] and oncolytic parvovirus in cervical and pancreatic carcinomas [42]. The CAL51 and MDA-MB-436 cells were the most sensitive to adenoviral infection and to virus- and HDACi-induced cell killing. However, despite the efficient virus-drug-mediated cell killing, replication in the presence of TSA and Scriptaid was either not enhanced or was reduced, suggesting that rapid cell elimination prevented potent viral production and spread in the more sensitive cells. The increased cell killing paralleled by decreased replication was significant at higher drug doses (≥0.5 µM), while an overall more than additive oncolytic effect was achieved with low concentrations of virus and drugs. By optimising both the timing and the dose of the inhibitor, viral efficacy was improved, including higher levels of replication. These data agree with previous reports proving the requirement for temporal histone- and protein-acetylation for optimal adenovirus propagation [16,17,18,19]. Our results show that TSA stimulates viral replication and cell killing when combined with Ad∆∆ and Ad-3∆-A20T at low doses. Both the benzamide MS-275 and the cyclic tetrapeptide, Romidepsin that target HDAC1, 2 and 3 [32] also supported Ad∆∆-mediated oncolysis. TSA and Romidepsin has been shown to induce apoptosis in combination with Ad-mediated p53 gene therapy in BCa cells [43], and to enhance gene expression of the OAd OBP-301 in renal and lung cancers [44,45]. We showed that simultaneous addition with virus of all three inhibitors, MS-275, Romidepsin and Scriptaid, mimicked the TSA-mediated effects by supporting viral infection and replication in MDA-MB-436, CAL51 and SUM159 cells.

E1A is the first viral protein expressed after infection and is essential for inhibiting Rb-family proteins to induce S-phase in normal cells [19]. One essential function for viral propagation to proceed is E1A binding to p300/CBP that removes the complex from active promoters and relocates the complex to acetylate histone tails on promoters regulating genes that promote cell cycle entry and progression in non-cancerous cells [16,17]. The association of cellular histones (mainly H3) with the viral DNA modulates the gene expression at specific stages of the viral cycle. For example, the interaction of E1A proteins with the p300/CBP lysine acetyltransferases causes a significant overall reduction of H3K18ac levels at the majority of cellular promoters but increases H3K18ac levels at promoters of genes involved in cell cycling and DNA replication [19]. It is likely that HDACis, including TSA and Scriptaid, which prevent deacetylation, support higher levels of acetylation at promoters that are essential for viral propagation, such as those regulating the cell cycle pathways. A global increase in histone acetylation may support the initial early viral gene expression and DNA amplification to initiate potent virion production, but not the later stages of virion assembly and release. Further studies would be necessary to delineate the lysine acetylation patterns on cellular and viral promoters in response to each HDACi and the respective effects on the viral life cycle. It was previously demonstrated that HDAC2 interacts with E1A and has a key role for the activation of gene expression, which cannot be substituted by HDAC1 and HDAC3 [46]. Our results demonstrate that Class I HDACs have important roles in adenovirus function and propagation also in cancer cells. Even though Scriptaid preferentially inhibits Class I HDACs, numerous additional cellular pathways are affected by both TSA and Scriptaid. This broad specificity is likely contributing to the attenuated viral replication in the presence of high concentrations of the inhibitors.

The TNBC and PDAC cell lines in our study carry highly diverse genetic alterations and originate from various tumour sites. For example, CAL51 and MDA-MB-436 are derived from metastatic tumours (pleural effusion), while the origin of SUM159 is a primary tumour. MDA-MB-436 and CAL51 do not express PTEN, a tumour suppressor gene that has a key role in avoiding the transformation of normal cells [47]. One of the reasons why PTEN is not expressed in human cancers is gene silencing, which could be caused by deregulated expression of HDACs in cancer, and HDACis may reactivate PTEN. In addition, MDA-MB-436 has a mutation in the p53 tumour suppressor gene, contributing to deregulated cell death and proliferation. The SUM159 cells are mutated in the PIK3CA pathway and have carcinosarcoma features that make these cells especially refractory to treatment [48]. The typical deregulated pathways in PDAC include a dysfunctional p53 pathway and constitutively activate KRAS, which also enable efficient cell killing with Ad∆∆ and enhanced viral replication in combination with TSA. Our findings demonstrate that cancer cells with various deregulated pathways are efficiently killed by combining Ad∆∆ or Ad-3∆-A20T, with HDACis targeting Class I enzymes as well as pan-inhibitors including Class IIb. The diverse genetic alterations in the tested TNBC and PDAC cells contribute to the observed differences in optimal time for the addition of inhibitors in combination with OAds. One reason for this may be the rate of viral gene expression or virion assembly in each cell type.

These combination strategies may offer a possible advance for future treatments of both TNBC and PDAC after in depth studies of treatment schedules in animal models and early phase clinical trials. Combining HDACis with tumour-targeted OAds may be a promising strategy to overcome current limitations in treatment resistance, delivery to metastatic distant lesions and intratumoural spread of virus.

## 5. Conclusions

The data provide evidence that HDACis increase the potency of Ad∆∆ and Ad-3∆-A20T to infect, replicate and kill TNBC and PDAC cells. Interestingly, both the pan-HDACi TSA and the Class I selective HDACi Scriptaid significantly improved viral functions, suggesting that simultaneous inhibition of several HDACs affect the viral life-cycle in cancer cells. The enhancement in viral activity was dependent on the specific cell line and presumably the specific gene alterations, reflected in different responses to timing and dose of inhibitor treatment. We show that HDACs play an important role in regulating OAd activity in cancer cells for both Ad∆∆ and the retargeted Ad-3∆-A20T mutants. Our data also demonstrate that the combinations cause efficient tumour growth inhibition in an aggressive TNBC xenograft model in vivo and that early viral genes (E1A) are potently expressed in the presence of TSA in Suit2 xenografts in vivo. We found that both Scriptaid and TSA enhance viral cancer cell killing, while the more specific inhibitor Scriptaid was more effective in increasing viral replication and thus the virus may spread more efficiently within the tumour tissue. Importantly, the data presented here warrants further investigation of specific cellular targets and mechanisms of action for future translation into promising clinical therapies for TNBC and PPDAC patients.

## Figures and Tables

**Figure 1 viruses-14-01006-f001:**
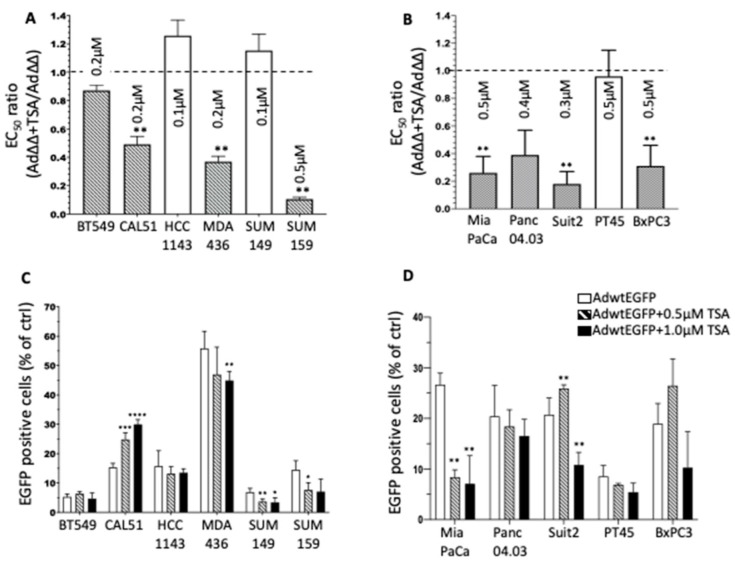
The pan-HDACi Trichostatin A (TSA) promotes adenovirus-mediated cell killing and infection in triple negative breast cancer (TNBC) and pancreatic ductal adenocarcinoma (PDAC) cell lines. (**A**,**B**) Ratios of EC_50_ for Ad∆∆ + indicated TSA concentrations (µM) relative to EC_50_ for Ad∆∆-infected cells alone, generated from dose-response curves to virus in cell viability assays (6 d), TNBC (**A**) and PDAC cells (**B**), dashed columns indicate sensitisation by TSA vs. AAd∆∆ alone, ** *p* < 0.01. (**C**,**D**) Cells were infected at 100 ppc with Ad5wtEGFP ± TSA (0.5 and 1 µM) added simultaneously, and cells were analysed by FACS 48 h later, TNBC (**C**) and PDAC (**D**) cells. Data are expressed as averages ± standard deviation (SD). * *p* < 0.05; ** *p* < 0.002; *** *p* < 0.0002; **** *p* < 0.0001; compared to Ad∆∆ infected cells.

**Figure 2 viruses-14-01006-f002:**
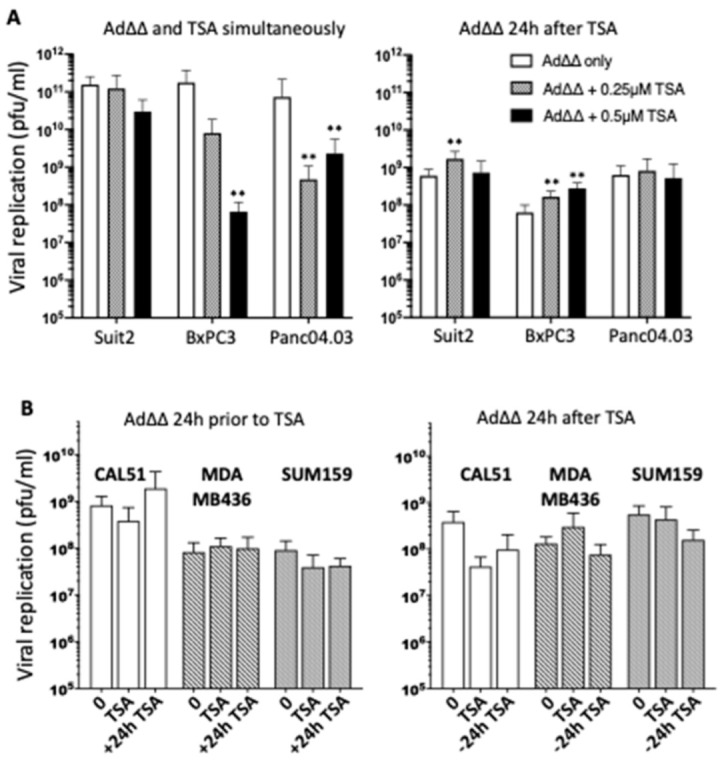
Efficient Ad∆∆ replication in PDAC cells and TNBC cells and in vivo efficacy in SUM159 tumour xenografts. (**A**) PDAC cells were infected with Ad∆∆ at 100 ppc and treated with 0.25 µM or 0.5 µM TSA. Left panel: TSA was added simultaneously with infection. Right panel: TSA was added 24 h prior to infection, ** *p* < 0.002. (**B**) TNBC cells were infected with Ad∆∆ at 100 ppc and 0.25 µM TSA was added; cells and medium were harvested 48 h after infection. Left panel: TSA was added 24 h after infection (+24 h TSA) and compared to cells treated simultaneously (TSA) or without TSA (0). Right panel: TSA was added 24 h before infection (−24 h TSA), and compared to cells treated simultaneously (TSA) or without TSA (0), *p* > 0.05. (**C**) Relative Ad∆∆ (100 ppc) replication rates over time when TSA was added under optimal conditions in each cell line. CAL51 and SUM159; TSA added 24 h after infection, MDAMB436; TSA added simultaneously with virus. Infectious units (pfu/mL) were determined by TCID50 assays and expressed as percentages of the respective treatment at 24 h. (**A**–**C**) Data expressed as averages ± SD, * *p* < 0.05, ** *p* < 0.002, *n* = 2. (**D**,**E**) SUM159 xenografts in athymic mice treated with Ad∆∆ at 1 × 10^10^ vp on day 1, 3 and 7 intratumourally or 1.5µg TSA/g on day 2, 4 and 8 intraperitoneally, or combined treatment with Ad∆∆ + TSA at the same dose schedule. Tumour volumes shown for day 15 and 21 after last day of treatment (**D**) and time to progression (**E**) (<500 mm^3^; * *p* < 0.05 (Ad∆∆ + TSA vs. Ad∆∆ and TSA alone), 6 animals/group.

**Figure 3 viruses-14-01006-f003:**
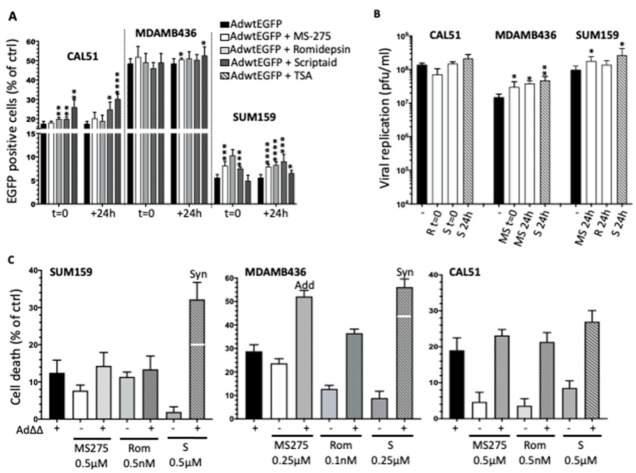
Selective HDACis promote adenovirus-mediated infection, replication and cell killing in TNBC cell lines. (**A**) Percentages of cells expressing EGFP after infection with AdwtGFP (100 ppc) and treated with MS-275 (0.1 µM; CAL51, 0.5 µM; MDA436 and SUM159); Romidepsin (0.5 nM; CAL51 and 0.1 nM; MDA-MB-436 and SUM159); Scriptaid (0.25 µM; CAL51 and 0.5 µM; MDA436 and SUM159); and TSA (0.25 µM for all cell lines), added simultaneously (t = 0) and 24 h post infection (+24 h). Cells were analysed by FACS 48 h post-infection. Data are expressed as averages of two different experiments performed in triplicates ± standard deviation (SD). (**B**) Cells were infected with Ad∆∆ at 100 ppc and treated with Romidepsin (0.1 nM at t = 0; CAL51 and 0.5 nM at +24 h; SUM159); Scriptaid (0.1 µM at t = 0 and +24 h; CAL51, 0.5 µM at +24 h; MDA-MB-436 and SUM159); and MS-275 (0.5 µM t = 0 and +24 h; MDA-MB-436 and SUM159). Cells and medium were harvested 48 h after infection. Data expressed as averages ± SD, *n* = 2. (**C**) Cell viability in infected SUM159 (Ad∆∆; 0.1 ppc), MDA-MB-436 (Ad∆∆; 4 ppc) and CAL51 (Ad∆∆; 0.2 ppc) treated with the indicated doses of each HDACi 24 h post-infection followed by analysis of cell death 6 d post-infection. Data expressed as % of dead cells relative to uninfected and untreated cells, *n* = 2. Synergistic cell killing is indicated (Syn) and correspond to levels above the theoretical additive values (Add), indicated by white line. A and B) Data expressed as averages ± SD. **** *p* < 0.0001, *** *p* < 0.001, ** *p* < 0.01, * *p* < 0.05.

**Figure 4 viruses-14-01006-f004:**
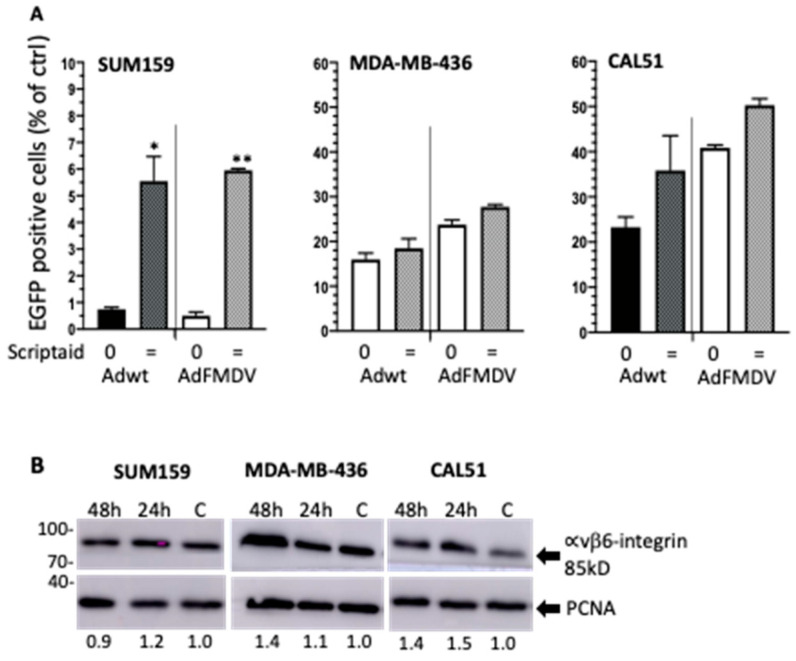
The TNBC cell lines express the cancer-specific αvß6-integrin and support high levels of infection with the retargeted Ad-3∆-A20T. (**A**) Cells were infected with 100 ppc of Ad5wtGFP and Ad-3∆-A20T-EGFP viruses alone (0) or in combination with 0.5 µM Scriptaid added simultaneously with virus (=). GFP expression was determined 48 h post-infection, *n* = 2. Data expressed as averages ± SD. ** *p* < 0.01, * *p* < 0.05. (**B**) Detection of αvß6-integrin (85 kDa) and PCNA (29 kDa), post-addition of 0.25 µM (CAL51) or 0.5 µM Scriptaid (MDA-MB-436 and SUM159). Control cells (no drug) was harvested after 48 h and drug-treated cells 24 h and 48 h after treatment, 25 µg protein/lane. Bands were quantified (ImageJ) and normalised to PCNA and untreated cells (indicated below blots).

**Figure 5 viruses-14-01006-f005:**
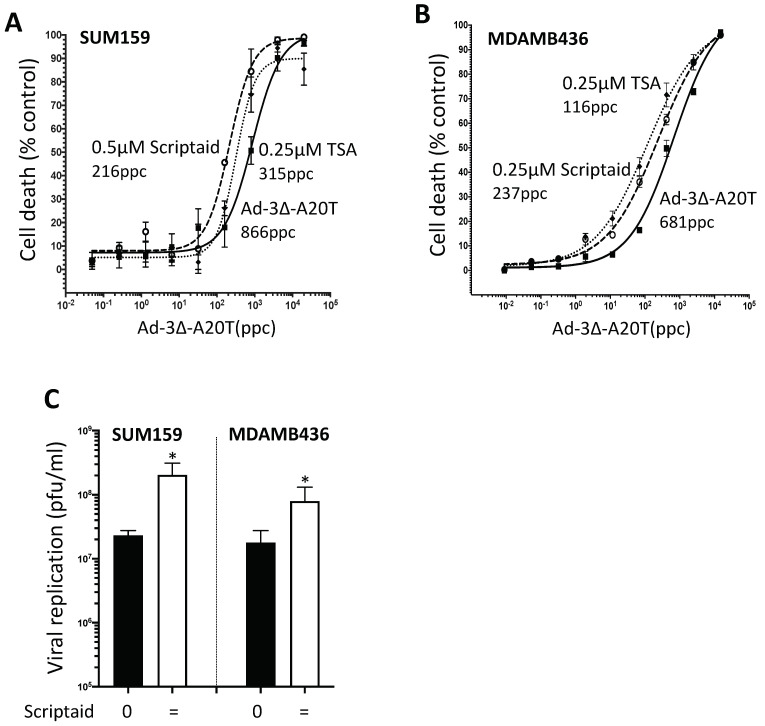
Addition of Scriptaid or TSA supports viral replication and sensitisation to the Ad-3Δ-A20T mutant in TNBC cells. SUM159 (**A**) and MDA-MB-436 (**B**) cells were infected with Ad-3∆-A20T at increasing concentrations and treated with Scriptaid at 0.5 µM (SUM159) or 0.25 µM (MDA-MB436) or 0.25 µM TSA added simultaneously. Cell viability was determined by MTS assay 3 d post-infection and EC_50_ values calculated, one representative study out of three, in triplicate. (**C**) SUM159 and MDA-MB436 cells were infected with Ad-3∆-A20T and treated with 0.1 µM Scriptaid simultaneously. Cells and medium were harvested 48 h post-infection and assayed by TCID50 for replication. Data are expressed as averages ± SD (*n* = 2, in duplicates), * *p* < 0.05.

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
