# Peer review of "HDAC Inhibitors Enhance Efficacy of the Oncolytic Adenoviruses Ad∆∆ and Ad-3∆-A20T in Pancreatic and Triple-Negative Breast Cancer Models"

_viruses, 2022, doi:10.3390/v14051006_

Round 1

Reviewer 1 Report

Authors demonstrate in this work that histone deacetylase inhibitors (HDACi) potently enhanced viral replication for Oncolytic adenoviruses (OAds), improving cell killing on a treatment time and dose depending manner and with clinical implications. Although the work is very original and interesting a few points should be risen:

•    Authors apply their strategy to breast and pancreatic tumours. Is there any specific relation on the acetylation status on those type of tumours? or could it be applyied to other types of neoplasias?. 
•    pRb/p53 might be important pieces on the differences between cell lines. It would be desirable a more extense description of pRb/p53 involvement in this strategy and how the deletion on the binding regions works on the viral replication. 
•    SUIT2 xenografts are treated with 10 exp9 vp but SUM159 xenografts were treated with ten times more of viral doses. Could authors explain what are those differences based on?
•    Could it be posible that the differences on the treatment sensitivity on the different cell lines could be due to the pRB/p53 status?. A table with known mutations for each cell line would be important. 
•    What would be the future strategy to be applied on the clinic?. Would the drug and virus be encapsulated together? Could you mention the advantages or disadvantages in the discussion?
•    Fig 2 A, B were only studied at 24h. For those type of studies, it seems a bit short. How authors think that the effect could it be for longer timeframes?? Why did they choose those time points??
•    Fig 2 D, E, F contain data from different cell types. It would be desirable to show the whole figure related to the same tumour type. Other cells lines should be located in supplemental information. 
•    Units on microliter to measure tumour size is uncommon (line 187 for example). Could authors change the units to 2/3 D measures, as cm/mm??
•    Although discussion is wide and nicely explained, a better description of how acetylation/deacetylation Works and differences between compounds used here would be appreciated.
•    Line 77: Administration route for Suit 2 xenografts was not described. 
•    Line 97: What are the increasing doses mentioned by the authors??
•    Line 248: No units were indicated

Reviewer 2 Report

This is a nice paper with importance and relevance to the potential clinical translational of oncolytic viruses in combination with sensitising agents. The study is thoughtful and thorough, and build on previous pioneering work from this group on their elegant selective virotherapy Ad5-3delta-A20T which has significant clinical promise, together with work from previous groups and studies looking at combinations of HDACi and oncolytic adenoviruses (e.g. Hulin-Curtis et at, 2018). The data appear to be clear and important and the manuscript conveys some important messages that may be critical for onward translation of such agents in the future. I have some comments which might be taken on board by the authors in a revision.

  1. The putative interaction between the fiber knob domain and blood clotting factors (FIX) is a red herring, that has even been acknowledged by those who originally presented this work (the interaction was first described here https://pubmed.ncbi.nlm.nih.gov/15919903/ then was shown not to occur by the sae authors here https://pubmed.ncbi.nlm.nih.gov/18391209/ (supplemental biacare figure 1). The authors might therefore include in the introduction and discussion how binding of FX (and it's ablation) might further enhance efficacy by limiting hepatotropism, especially when targeting intravenously (this is less of an issue for the direct intratumoral approach used here, but may still be relevant where leakage occurs into the blood).
  2. For the cell lines used, it is unclear what the origin of PT45 cells was.
  3. The methods mention in vivo studies were performed in both SUM159 and SUIT2 models, but I can only see SUM159 in vivo model. Why was one cancer type treated with 10x higher dose than the other (assuming both will be shown?).
  4. Figure numbers do not need to be mentioned on the figures themselves - they are available in the legend and hence do not need a title on the figure themselves. Figure 2 has "Figure 2" written on it twice.
  5. In figure 1 C+D, does GFP expression correlate directly with CAR expressed on these cell lines (ie the expression of the primary Ad5 receptor?).
  6. Figure 2 shows nicely important data that may be important for dosing strategies and the sequential timing of virus: HDACi treatment. My feeling is that the in vivo study might be better presented as a separate figure to break this up a little. 
  7. Why was the in vivo study stopped at day 29 when treatment appeared to be working?
  8. In figure 4A, the authors state n=2, so are error bars (and statistics) appropriate? I am not a statistician but I do not this this is appropriate for such low n numbers, and this should be increased. 
  9. Generally it would be good if either the figures could be enlarged or at least the font size could be increase. Some axes are very hard to read (e.g. figure 2C).

All in all this is a nice paper and an important one for the field, so I hope the reviewers will be able to make the minor adjustments suggested and improve the manuscript for publication.
